# The influence of the COVID-19 pandemic on the short- and long-term interactions in the agricultural market: Evidence from a connectedness network approach

**Jung-Bin Su** [ID]*

Department of Finance, School of Finance, Fuzhou University of International Studies and Trade, Fuzhou City, Fujian Province, China

* jungbinsu@gmail.com, jungbinsu@yahoo.com.tw, 180466672@qq.com

**Data Availability Statement:** All relevant data are within the paper and its Supporting information files.

## Abstract

This study employs a bivariate GARCH model to examine the influence of the COVID-19 pandemic on the interactions of the commodities in the agricultural market via a connectedness network approach. Empirical results show that this pandemic alters the commodities' roles—the activators, net transmitters, and net receivers—in the volatility and return connectedness but not for the activators in the correlation connectedness. Moreover, this pandemic enhances the interactive degree of the unidirectional negative return spillovers and the bidirectional distinct-sign volatility spillovers but doesn't for the interactive degree of correlation. Thus, the COVID-19 pandemic, a short-term drastic event, can influence short-term interactions like volatility and return spillovers but can't affect one long-term interaction like the correlation. Nevertheless, this pandemic raises the intensity of the correlation as well as volatility and return spillovers. The findings provide policymakers to make short- and long-term investment strategies in the agriculture market.

## 1. Introduction

Everybody knows that food is a crucial article to survive for human beings. However, climate change such as sudden floods or droughts has made agricultural commodities like grains be planted not smoothly in recent years. Moreover, in 2020, the farmers and workers were infected by the coronavirus disease 2019 (COVID-19) pandemic. To reduce the spread of the virus, the governments in many countries enacted a series of restrictive policies such as the social distancing policy and lockdown measures because social distancing lowered the average daily infection cases by 12% in the United States [1, 2]. These restrictive policies altered the mode of management in financial institutions. For example, the demand for fintech and traditional bank loans increased at the aggregate and individual levels after the pandemic, especially for fintech loans [3]. Moreover, the above policies also made agricultural products from planting to transportation run not swimmingly [4, 5]. For instance, the lockdown measures resulted in logistics disruption because of the interruption to transportation, and this disruption further caused a series of problems in agricultural production such as price rise,

**Funding:** The author(s) received no specific funding for this work.

**Competing interests:** The authors have declared that no competing interests exist.

production means shortage, sales reduction, and, finally, rural households' losses [6, 7]. These phenomena have resulted in worrying about food shortages for people and further prompted the food crisis issues to be paid attention to worldwide [8–10]. Additionally, in the past years, globalization has driven financial integration and liberalization of trade and investment between economies. This causes the trend of the price level of different commodities to be related more closely [9, 11, 12]. However, climate change and the COVID-19 pandemic are long-term and short-term factors of the food crisis issue, respectively. This motivates us to examine the influence of the COVID-19 pandemic on the interactions among the commodities in the agricultural market to propose some long- and short-term policies for the government's agricultural sector in response to this short-term extreme event. According to the methods to inspect the interaction between two commodities, the past literature was divided into two categories.

The first category utilized the significant situation on parameters related to the long- and short-term interactions in a multivariate GARCH model to examine the interactions between the commodities in the agriculture market and other markets. For instance, Mensi et al. inspected the volatility and return spillovers between barley, corn, sorghum, and wheat in the agriculture market and WTI and Brent crude oil, gasoline, and heating oil in the energy market [9]. They found that there exists a bidirectional spillover effect across barley and each of the crude oil and gasoline markets. Moreover, the price level of all commodities has been managed by dynamic conditional correlations with a common enlarging inclination during the global financial crisis (GFC) in 2008. Moreover, Chang et al. discovered that volatility spillovers subsist in all four kinds of financial assets (futures, spot, financial index, and ETF) in three different commodities- ethanol in the energy market and corn and sugar in the agriculture market [13]. Similarly, Han et al. showed that a bidirectional volatility linkage exists between the agricultural market from corn, soybean, and wheat and the energy market from WTI crude oil and natural gas [14].

Sadorsky found that the dynamic conditional correlations between the stock and each of wheat, oil, and copper increased between 2008 and 2009 [15]. Ahmed and Huo discovered that a unidirectional shock spillover exists from the stock market to most commodity markets like soybean, wheat, gold, and copper [16]. Garcia-Jorcano and Sanchis-Marco found that the highest spillovers from oil and the US commodity index to wheat existed in volatile periods like the post-Draghi speech and COVID-19 periods [17]. Similarly, Liu et al. discovered that a weak correlation exists in the standard period, whereas this correlation intensifies and becomes more complicated during the COVID-19 era. Moreover, during the COVID-19 outbreak, bidirectional return and volatility spillovers between stock-commodity markets are more outstanding [18].

The second category used the net spillover index proposed by Diebold and Yilmaz based on the time-varying parameter vector autoregressive model with generalized forecast error variance decomposition to inspect the interactions between the commodities in the agriculture market and other markets [19, 20]. For instance, Dahl et al. found bidirectional volatility spillover among the futures markets of agricultural commodities and crude oil intensifies during financial and economic turmoil periods [8]. Moreover, net volatility spillover enhanced in periods of huge falls in crude oil prices in 2008 and later in 2014. As shown in a network by [21], rice and crude oil markets are receivers, even if the role of rice is significantly reduced on all time scales after the COVID-19 pandemic. Similarly, in the volatility spillover networks in [22], we found both the COVID-19 pandemic and the Russia-Ukraine conflict have led to increased connectedness, whereas the pandemic has been more significant. Furthermore, fossil energy and 'wheat and corn' were the main risk transmitters in the early period of the COVID-19 pandemic and the period of the Russia-Ukraine conflict, respectively. Wang found

the return spillover and dynamic linkage varied with time and were easily affected by major crises such as the COVID-19 pandemic [23].

Kang et al. found that bidirectional behavior subsisted in volatility and return spillover indices across commodity futures markets. This tendency is more outstanding in the aftermath of recent financial crises [12]. Mensi et al. also discovered that total spillovers had intensified during the economic crises in the US and China and the 2005 commodities crisis [10]. Similarly, Nekhili et al. also showed that the major events, including the oil price crash, the GFC in 2008–2009, the European sovereign debt crisis, and the COVID-19 pandemic, intensified the considered markets' spillover effects [24]. Moreover, Dai and Zhu discovered that total volatility spillover has a huge rise during major crisis events such as the COVID-19 pandemic [25]. Additionally, Farid et al. also found that during this pandemic, there exists a strong transmission of return shocks between metal, energy, and agricultural commodities [11]. Hence, they found that the spillovers among commodities are time-varying and crisis-sensitive, and these spillovers are always intensified by economic and political events.

To sum up, we get the following conclusions from the literature review. Firstly, we found that the multivariate GARCH model in the first category of literature can determine whether the spillover between two assets is significant or not and whether this spillover is negative or positive. However, they never investigated the role played by each asset in a group of assets—a net receiver or net transmitter in a system. On the other hand, the net spillover index in [19] in the second category of literature can find the role played by each asset in a group of assets: a net receiver or net transmitter in a system. However, they can't own the functions in the first category of literature: whether the spillover is significant or not and whether the spillover is positive or negative. Secondly, the high and more significant return and volatility spillovers subsisted in the volatile rather than the tranquil period. Moreover, the intensity of short- and long-term interactions increases in the volatile period. The volatile period always appears after economic, political, or other drastic events, such as the GFC in 2008 and the COVID-19 pandemic in 2020.

To fill the crevice in the past literature, we embed the concept of net transmitter and net receiver found by the net spillover index of [19] into the method of multivariate GARCH model. Accordingly, this study uses seven commodities that often appear in the literature related to the agricultural market- wheat, corn, oat, soybean, soybean oil, coffee, and sugar- as the data. Notably, soybean and soybean oil are the major raw materials of biodiesel and renewable diesel [26]. More importantly, this study mainly uses the connectedness networks in the agricultural market to describe the interactions between alternative two assets within a group of commodities in the agricultural market from the view of the significance of the interaction. Moreover, the return series of most financial assets displays linear dependence and a strong autoregressive conditional heteroskedasticity (ARCH) effect [27–30]. The above facts indicate that the bivariate GARCH model is suitable because this model can seize the volatility heterogeneity or volatility changing over time in the financial return series. In addition, this study aims to explore the influence of the COVID-19 pandemic on short- and long-term interactions. Hence, we add two time-dummy variables to the above model to explore the interaction between two commodities before and after the epidemic, so the bivariate GARCH model with two time-dummy variables is very suitable for this research topic.

Then, for a pair of agriculture commodities, we employ a bivariate BEKK-GARCH model including two time-dummy variables to estimate the parameters related to two short-term and one long-term interactions in the pre-pandemic and post-pandemic periods. The above BEKK-GARCH model is derived by [27] by applying the recommendation of [31] to get a positive definite type of bivariate GARCH model with diagonal representation. Moreover, the BEKK model is named after Baba, Engle, Kraft, and Kroner [32]. Additionally, one long-term

interaction is the correlation but two short-term interactions are the volatility and return spillovers. Subsequently, we transfer the significant situations of the above parameters into the symbolic results to plot six network diagrams corresponding to three types of interactions for these two subperiods as inspired by the connectedness network approach in [10, 11, 22, 25]. The six network diagrams are used to depict, in the pre-pandemic and post-pandemic periods, the three types of interactive relationships between any two commodities within a group of agricultural commodities. In each of the six network diagrams, there are seven nodes to represent seven agricultural commodities such as wheat, corn, oat, soybean, soybean oil, coffee, and sugar. This is the **first** contribution in this study because instead of the spillover index of [19] used in [10, 11, 22, 25], this study is the first article to use the significant situations of parameters connected with the interactions to depict the interactive relationship between two assets denoted by two nodes in a network.

Regarding each node in a network, we propose four types of calculations to count the total number of arrows away from that node and point to that node as well as the summation and difference for the above two total numbers. The total number of arrows away from (respectively, point to) a specific node is similar to 'the total directional connectedness to (respectively, from) others' in the spillover framework of [19]. Hence, the difference between the above two total numbers is similar to 'the net total directional connectedness' in [19] because the net total directional connectedness is obtained by subtracting 'the total directional connectedness **to** others' from 'the total directional connectedness **from** others'. Hence, if the value of the above difference for this specific node is greater (respectively, less) than zero, then the asset corresponding to this node is a net transmitter (respectively, receiver) in a system. The results of four types of calculations are used to investigate the following two questions. Among a group of commodities, which commodity is the net receiver or net transmitter in the return spillover (or volatility spillover) network, and which commodity is the most or least active in the network of correlation and return and volatility spillovers? This is the **second** contribution in this study because instead of the net spillover index of [19] used in [10], we use the difference for the two total numbers of arrows away from that node and point to that node to determine the commodity corresponding to that node is a net receiver, a net transmitter, or a neutral in a network. In addition, we use the summation for the above two total numbers to determine which commodity is the most or least active in a group of assets denoted by the nodes in a network. The most active asset in a group of assets is called an activator in a network. The obtained results are used to make short- and long-term operation strategies for the fund managers, investors, and government officials to achieve the goal of risk diversification for this short-term extreme event.

Consequently, this study proposes the following seven hypotheses to inspect the influence of the COVID-19 pandemic on the interactions for a pair of agricultural commodities. This is the third contribution because, to the best of my knowledge, this work is the first article to examine the influence of the COVID-19 pandemic on the interactions between agricultural commodities from the viewpoints of net transmitters, net receivers, and activators in the connectedness networks. The following two subperiods represent the pre-pandemic and post-pandemic periods.

Hypothesis 1 (respectively, Hypothesis 4) is that the activators in the return (respectively, volatility) spillover network for the two subperiods are different.

Hypothesis 2 (respectively, Hypothesis 5) is that the net transmitters in the return (respectively, volatility) spillover network for the two subperiods are different.

Hypothesis 3 (respectively, Hypothesis 6) is that the net receivers in the return (respectively, volatility) spillover network for the two subperiods are different.

Hypothesis 7 is that the activators in the correlation network for the two subperiods are different.

The main findings can be summarized as follows. Firstly, in the return spillover network, coffee and wheat respectively are the activators in the pre-pandemic and post-pandemic periods. Coffee and wheat are the net transmitters in pre-pandemic but soybean oil, corn, and sugar for the post-pandemic period. Oat and corn are the net receivers in the pre-pandemic period but oat and soybean for the post-pandemic period. Additionally, corn is the net receiver in the pre-pandemic period but it is changed into the net transmitter in the post-pandemic period. Thus, the COVID-19 pandemic alters the commodities' roles in the return connectedness, indicating that this pandemic influences the return connectedness of commodities in the agriculture market. In addition, this pandemic enhances the interactive degree of unidirectional negative return spillovers and their intensity. Notably, soybean and soybean oil are the neutrals in the pre-pandemic period but the net receiver and net transmitter in the post-pandemic period, respectively.

Secondly, in the volatility spillover network, soybean and oat respectively are the activators in the pre-pandemic and post-pandemic periods. Corn, oat, and coffee are the net transmitters in the pre-pandemic period but wheat, oat, and soybean oil for the post-pandemic period. Soybean, wheat, and soybean oil are the net receivers in the pre-pandemic period but soybean and coffee for the post-pandemic period. Additionally, wheat and soybean oil are the net receivers in the pre-pandemic period but they are altered into net transmitters in the post-pandemic period. Moreover, coffee is the net transmitter in the pre-pandemic period but it switches into the net recipient in the post-pandemic period. Thus, the COVID-19 pandemic also changes the commodities' roles in the volatility connectedness. This proves that this pandemic influences the volatility connectedness of commodities in the agriculture market. Additionally, the COVID-19 pandemic raises the interactive degree of the bidirectional spillovers having positive and negative signs in different directions and it also enlarges their intensity. Notably, soybean and soybean oil are the net receivers in the pre-pandemic period whereas soybean oil is changed into the net transmitter in the post-pandemic period but doesn't change for soybean.

Thirdly, in the correlation network, the activators in the pre-pandemic and post-pandemic periods are corn, soybean, and sugar and they are completely the same. Thus, the commodities' role in the correlation connectedness isn't altered by this pandemic and this pandemic does not influence the correlation connectedness of commodities in the agriculture market. In addition, the commodities in the agriculture market have positively interactive relationships and this pandemic increases the intensity of correlation. To sum up, in the agriculture market, the COVID-19 pandemic influences the volatility and return connectedness of commodities but doesn't for the correlation connectedness, indicating that a short-term drastic event like this pandemic can influence two short-term interactions like the return and volatility spillovers but can't for the long-term interaction of correlation. To sum up, in the agriculture market, the COVID-19 pandemic influences the volatility and return connectedness of commodities but not the correlation connectedness. This indicates that a short-term drastic event of this pandemic influences two short-term interactions but not one long-term interaction. Moreover, soybean oil increased its importance in the agricultural market during the post-pandemic period attributed to its recent application in alternative fuels.

The remnant of this paper is arranged in the following sections. Section 2: Methodology illustrates a multivariate GARCH model used in this study. The descriptive statistics of the return series for the study data are reported in Section 3: Data and descriptive statistics. Section 4: Empirical results illustrates the results of the empirical model and investigates the topics proposed in this work. Finally, in section 5: Conclusion and further discussion we summarize

the findings in Data and descriptive statistics and Empirical results sections and give some discussion for them. Then, several policy implications are proposed for government officials, investors, and fund managers.

## 2. Methodology

This study examines the influence of the COVID-19 pandemic on the interactions between the commodities in the agricultural market via a connectedness network approach. Thus, a bivariate GARCH model including two time-dummy variables is employed to capture the behaviors of short- and long-term interactions in the two subperiods.

### 2.1 The specification of a bivariate GARCH model including two time-dummy variables

The mean equation of this model is designed as a bivariate vector autoregressive with lag 1 (hereafter, VAR (1)) to seize the return spillover for a pair of assets. On the other hand, the variance-covariance equation of this model is represented as a bivariate BEKK-GARCH (1,1)-X model with the normal distribution to capture the volatility spillover and correlation for a pair of assets. Hence, the variance-covariance equation ($H_t$) and mean equation ($r_t$) are two-dimension and we call this model the bivariate diagonal VAR (1)-BEKK-GARCH (1,1)-X model (hereafter, B-GARCH).

The two-dimensional mean equation, $r_t$, is listed below.

$$r_{1,t} = \phi_{10} + \phi_{11}r_{1,t-1} + \phi_{12}r_{2,t-1} + \varepsilon_{1,t}, \tag{1}$$

$$r_{2,t} = \phi_{20} + \phi_{21}r_{1,t-1} + \phi_{22}r_{2,t-1} + \varepsilon_{2,t}, \tag{2}$$

where $r_t = (r_{1,t}, r_{2,t})'$ represents a return column vector. $r_{i,t} = (\ln P_{i,t} - \ln P_{i,t-1}) \times 100$ and i = 1,2. $P_{i,t}$ denotes the $i^{th}$ asset's close price at time t for a pair of assets and $r_{i,t}$ is its corresponding return. '$\phi_{10}$, $\phi_{11}$, and $\phi_{12}$' and '$\phi_{20}$, $\phi_{21}$ and $\phi_{22}$' are two groups of parameters on the mean equations $r_{1,t}$ and $r_{2,t}$, respectively. If parameter $\phi_{ij}$ such as $\phi_{12}$ or $\phi_{21}$ is significant then a return spillover from the $j^{th}$ asset to the $i^{th}$ asset subsists in a pair of assets. $\varepsilon_t = (\varepsilon_{1,t}, \varepsilon_{2,t})'$ denotes a column vector of error terms. Moreover, its conditional distribution is presumed to obey a bivariate normal distribution with $E_{t-1}(\varepsilon_t) = 0$ and $E_{t-1}(\varepsilon_t\varepsilon_t') = H_t$. That is, $\varepsilon_t|\Omega_{t-1} \sim N(0, H_t)$.

Subsequently, the two-dimensional variance-covariance equation, $H_t$, is represented as follows.

$$h_t = vech(H_t) = [h_{11,t}, h_{12,t}, h_{22,t}]' \tag{3}$$

$$h_{11,t} = \omega_1 + \alpha_1\varepsilon_{1,t-1}^2 + \beta_1 h_{11,t-1} + \nu_{12}h_{22,t-1} \tag{4}$$

$$h_{12,t} = \omega_{12} + \alpha_{12}\varepsilon_{1,t-1}\varepsilon_{2,t-1} + \beta_{12}h_{12,t-1} \tag{5}$$

$$h_{22,t} = \omega_2 + \alpha_2\varepsilon_{2,t-1}^2 + \beta_2 h_{22,t-1} + \nu_{21}h_{11,t-1} \tag{6}$$

where vech ($H_t$) represents the vech operator that stacks the 'upper triangular' portion of a two-dimensional matrix $H_t$ into a vector with a single column. Moreover, $h_{11,t}$ and $h_{22,t}$ denote the variances of the first and second assets at the time t for a pair of assets, respectively. '$\omega_1$, $\alpha_1$, $\beta_1$, and $\nu_{12}$' and '$\omega_2$, $\alpha_2$, $\beta_2$, and $\nu_{21}$' are two groups of parameters on the variance equations $h_{11,t}$ and $h_{22,t}$, respectively. $h_{12,t}$ represents the covariance between the two abovementioned

assets' returns at time t for a pair of assets and $\omega_{12}$, $\alpha_{12}$, and $\beta_{12}$ denote the parameters on this covariance equation. If parameter $\nu_{ij}$ such as $\nu_{12}$ or $\nu_{21}$ is significant then there exists a volatility spillover from the $j^{th}$ asset to the $i^{th}$ asset in a pair of assets.

Notably, to seize the long- and short-term interactions for a pair of assets in the two subperiods, some parameters connected with the volatility and return spillovers as well as correlation should contain two time-dummy variables. Then, they are expressed as follows.

$$\phi_{12} = \phi_{12}^{B} \cdot d_t^{B} + \phi_{12}^{A} \cdot d_t^{A}, \ \phi_{21} = \phi_{21}^{B} \cdot d_t^{B} + \phi_{21}^{A} \cdot d_t^{A},$$

$$\nu_{12} = \nu_{12}^{B} \cdot d_t^{B} + \nu_{12}^{A} \cdot d_t^{A}, \ \nu_{21} = \nu_{21}^{B} \cdot d_t^{B} + \nu_{21}^{A} \cdot d_t^{A},$$

$$\omega_{12} = \omega_{12}^{B} \cdot d_t^{B} + \omega_{12}^{A} \cdot d_t^{A} \tag{7}$$

where $d_t^{B}$ and $d_t^{A}$ represent two time-dummy variables, which partition the study period into the pre-pandemic and post-pandemic periods that stand for the periods respectively before and after the date of the COVID-19 pandemic. $d_t^{B} = 1$ if $date_{start} \leq t < date_{covid19}$, and $d_t^{B} = 0$ otherwise; $d_t^{A} = 1$ if $date_{covid19} \leq t \leq date_{end}$, and $d_t^{A} = 0$ otherwise. $date_{start}$ (respectively, $date_{end}$) represents the start (respectively, end) date of the study sample. $date_{covid19}$ denotes the date of the COVID-19 pandemic occurring on March 11, 2020.

Additionally, via the maximum likelihood (ML) optimizing procedure, the parameters in this bivariate GARCH model are estimated by the following bivariate log-likelihood function with normal density.

$$L(\boldsymbol{\psi}) = \sum_{t=1}^{n} \ln\{f(\mathbf{r_t}|\Omega_{t-1}; \boldsymbol{\psi})\} = -\frac{n}{2}\ln 2\pi - \frac{1}{2}\sum_{t=1}^{n}\left(\ln|\mathbf{H_t}| + \boldsymbol{\varepsilon_t}\mathbf{H_t}^{-1}\boldsymbol{\varepsilon_t}\right) \tag{8}$$

where $\boldsymbol{\Psi} =$
$[\phi_{10}, \phi_{11}, \ \phi_{12}^{B}, \ \phi_{12}^{A}, \ \phi_{20}, \phi_{21}^{B}, \ \phi_{21}^{A}, \phi_{22}, \omega_1, \alpha_1, \beta_1, \nu_{12}^{B}, \nu_{12}^{A}, \omega_{12}^{B}, \ \omega_{12}^{A}, \alpha_{12}, \ \beta_{12}, \omega_2, \ \alpha_2, \beta_2, \ \nu_{21}^{B}, \nu_{21}^{A}]$ is the vector of parameters of this model. $\Omega_{t-1}$ represents the information set of all observed returns up to time $t - 1$ and $f(\cdot)$ represents the bivariate normal density. In addition, n is the sample size in the estimate period. $\mathbf{r_t}$, $\mathbf{H_t}$, and $\boldsymbol{\varepsilon_t}$ are shown in Eqs (1)–(6). Additionally, parameters with the superscripts 'B' and 'A' can seize the financial feature connected with that parameter in the pre-pandemic and post-pandemic periods, respectively. For instance, parameters $\phi_{ij}^{B}$ (respectively, $\phi_{ij}^{A}$) and $\nu_{ij}^{B}$ (respectively, $\nu_{ij}^{A}$) are applied to examine whether, in the pre-pandemic (respectively, post-pandemic) period, there exists a return and volatility spillovers from the $j^{th}$ asset to the $i^{th}$ asset, respectively.

## 2.2 The theory of constructing the connectedness networks

This study aims to explore the influence of the COVID-19 pandemic on short- and long-term interactions via a connectedness network approach. Additionally, the short-term interactions are the volatility and return spillovers, which are directional financial features. Conversely, the long-term interaction is the correlation, which belongs to the non-directional financial feature. Hence, in this subsection, we illustrate how to use the significance of parameters related to two short-term interactions ($\phi_{ij}$ and $\nu_{ij}$) and one long-term interaction ($\omega_{ij}$) to get the corresponding connectedness networks.

If parameter $\phi_{ij}$ (or $\nu_{ij}$) is significantly positive (negative) then a positive (negative) return (or volatility) spillover from the $j^{th}$ asset to the $i^{th}$ asset subsists in a pair of assets. Subsequently, in a network of return (or volatility) spillover, we utilize a red (blue) arrow from a node denoting the $j^{th}$ asset to another node denoting the $i^{th}$ asset to represent this positive (negative)

return (or volatility) spillover. Regarding each node in a return (or volatility) spillover network, we count the total number of arrows away from that node and point to that node as well as the summation and difference for the above two total numbers. The results of four types of calculations are recorded as four numbers in a bracket beside this node. That is, the first (second) number in this bracket records the total number of arrows away from (point to) that node. The third number in this bracket denotes the summation between the first and second numbers in the same bracket or records the summation for the above two total numbers. If this number in a specific node is the greatest (smallest) number among the third numbers for all nodes, then the asset corresponding to this specific node is the most (least) active asset in this network, regulating that the outer circle of this node is marked in solid (dash) line. The fourth number in this bracket denotes the difference between the first and second numbers in the same bracket or records the difference for the above two total numbers. If the difference is greater (less) than zero, then the asset corresponding to this node is a net transmitter (receiver) in a system and this node is marked in red (yellow). In addition, if the difference is equal to zero, then the asset corresponding to this node is neutral in a system and this node is marked in green.

If parameter $\omega_{ij}$ is significantly positive (negative) then a positive (negative) correlation between the $i^{th}$ asset and the $j^{th}$ asset exists in a pair of assets. Subsequently, in a network of correlation, we utilize a red (blue) line between a node denoting the $i^{th}$ asset and another node denoting the $j^{th}$ asset to represent this positive (negative) correlation. Regarding each node in a correlation network, we calculate the total number of lines connected with this node and record this number in a bracket beside this node. If this number in a specific node is the greatest (smallest) number among these numbers for all nodes, then the asset corresponding to this node is the most (least) active asset in this network, regulating that the outer circle of this node is marked in solid (dash) line.

## 3. Data and descriptive statistics

This work primarily employs a connectedness network method to inspect the influence of the COVID-19 pandemic on the interactions among the seven agricultural commodities. Hence, the study data contain daily close price data of wheat, corn, oat, soybean, soybean oil, coffee, and sugar in the agricultural market. The prices of wheat, corn, Oats, and soybean are measured by U.S. Dollars per bushel whereas those of soybean oil, coffee, and sugar are U.S. Dollars per pound. All data was downloaded from the website of https://www.macrotrends.net and they cover the period from March 26, 2013, to March 11, 2021. Consequently, according to the date the COVID-19 pandemic occurred on March 11, 2020, we partitioned the study data into the pre-pandemic and post-pandemic periods. Notably, Wheat, corn, oat, soybean, and soybean oil belong to the grains type of agricultural commodities. On the contrary, coffee and sugar are included in the beverage type of agricultural commodities. In addition, soybean oil is the raw material of renewable diesel and it is extracted from the soybean.

Panel A and Panels B-C of Table 1 report the descriptive statistics of the daily return of the study data in the overall period as well as its two subperiods. Subsequently, we execute two types of performance comparison for the values in columns 'Mean', 'SD', and 'Rᵃ' listed in Table 1. The first (respectively, second) type of performance comparison is to compare the values in columns 'Mean', 'SD', or 'Rᵃ' for three periods (respectively, seven commodities) based on the same commodity (respectively, period). Three periods are the overall period as well as the pre-pandemic and post-pandemic subperiods. Regarding the first type of performance comparison, we find that, for most commodities, the smallest (respectively, greatest) values of 'SD', 'Mean', or 'Rᵃ' are dispersed in the pre-pandemic (respectively, post-pandemic) period.

**Table 1. Descriptive statistics of daily return for the overall, pre-pandemic and post-pandemic periods.**

| | Mean | SD | R$^a$ | Max. | Min. | SK | KUR | J-B | Q$^2$ (24) |
|---|---|---|---|---|---|---|---|---|---|
| **Panel A. The overall period** | | | | | | | | | |
| wheat | -0.0061 | 1.636 | -0.0037 | 6.580 | -6.882 | 0.255[c] | 0.823[c] | 78.2[c] | 183.1[c] |
| corn | -0.0154 | 1.412 | -0.0109 | 7.891 | -7.929 | -0.258[c] | 2.699[c] | 630.0[c] | 171.6[c] |
| oat | -0.0020 | 1.798 | -0.0011 | 7.609 | -6.767 | -0.062 | 0.903[c] | 69.3[c] | 98.6[c] |
| soybean | -0.0008 | 1.198 | -0.0006 | 5.450 | -6.583 | -0.199[c] | 2.128[c] | 390.9[c] | 279.8[c] |
| soybean oil | 0.0039 | 1.211 | 0.0032 | 6.647 | -5.375 | 0.115[b] | 0.979[c] | 84.3[c] | 134.5[c] |
| coffee | -0.0012 | 2.028 | -0.0005 | 11.789 | -7.361 | 0.264[c] | 1.704[c] | 265.5[c] | 505.6[c] |
| sugar | -0.0046 | 1.746 | -0.0026 | 10.456 | -5.643 | 0.244[c] | 1.293[c] | 159.4[c] | 110.7[c] |
| **Panel B. The pre-pandemic period** | | | | | | | | | |
| wheat | -0.0189 | 1.631 | -0.0116 | 6.580 | -6.882 | 0.233[c] | 0.924[c] | 78.1[c] | 200.6[c] |
| corn | -0.0379 | 1.409 | -0.0269 | 7.891 | -7.929 | -0.292[c] | 2.955[c] | 661.0[c] | 169.8[c] |
| oat | -0.0215 | 1.823 | -0.0118 | 7.609 | -6.767 | -0.022 | 0.906[c] | 60.0[c] | 91.5[c] |
| soybean | -0.0283 | 1.207 | -0.0234 | 5.450 | -6.583 | -0.177[c] | 2.200[c] | 361.9[c] | 262.0[c] |
| soybean oil | -0.0343 | 1.167 | -0.0294 | 6.647 | -5.371 | 0.208[c] | 1.173[c] | 113.0[c] | 102.9[c] |
| coffee | -0.0097 | 2.013 | -0.0048 | 11.789 | -6.417 | 0.306[c] | 1.818[c] | 268.2[c] | 498.5[c] |
| sugar | -0.0203 | 1.703 | -0.0119 | 10.456 | -5.643 | 0.278[c] | 1.499[c] | 186.3[c] | 81.4[c] |
| **Panel C. The post-pandemic period** | | | | | | | | | |
| wheat | 0.0819 | 1.670 | 0.0490 | 5.129 | -4.053 | 0.394[b] | 0.173 | 6.89[b] | 28.37 |
| corn | 0.1404 | 1.426 | 0.0984 | 4.953 | -4.636 | -0.043 | 1.017[c] | 10.98[c] | 41.44[b] |
| oat | 0.1324 | 1.612 | 0.0821 | 4.061 | -5.228 | -0.395[b] | 0.788[b] | 13.15[c] | 34.37[a] |
| soybean | 0.1890 | 1.118 | 0.1689 | 3.278 | -4.363 | -0.318[b] | 1.623[c] | 32.06[c] | 54.19[c] |
| soybean oil | 0.2687 | 1.459 | 0.1840 | 3.429 | -5.375 | -0.420[c] | 0.380 | 8.98[b] | 33.30[a] |
| coffee | 0.0577 | 2.134 | 0.0270 | 6.861 | -7.361 | 0.005 | 1.109[c] | 12.97[c] | 24.37 |
| sugar | 0.1035 | 2.021 | 0.0512 | 5.624 | -5.361 | 0.050 | 0.317 | 1.17 | 63.16[c] |

Notes: 1. The superscripts a, b, and c on descriptive statistics denote that these descriptive statistics are significant at the 10%, 5%, and 1% levels, respectively. 2. Mean denotes the mean return whereas SD represents the standard deviation of return. R$^a$ denotes the realized risk-adjusted returns and it is obtained by the mean return divided by the standard deviation. 3. SK and KUR denote the skewness and excess kurtosis, respectively. 4. J-B statistics are based on [33] and are asymptotically chi-squared-distributed with 2 degrees of freedom. 5. Q$^2$ (24) statistics are asymptotically chi-squared-distributed with 24 degrees of freedom. 6. The bold and italic fonts in columns 'Mean', 'SD', and 'R$^a$' respectively denote the largest and smallest values of the mean, standard deviation, and the realized risk-adjusted returns when the values of mean, standard deviation, and the realized risk-adjusted returns for the overall, pre-pandemic, and post-pandemic periods are compared each other based on the same commodity. 7. The shade and underline fonts in columns 'Mean', 'SD' and 'R$^a$' respectively denote the largest and smallest values of the mean, standard deviation, and the realized risk-adjusted returns when the values of mean, standard deviation, and the realized risk-adjusted returns for all seven assets are compared each other based on the same period. 8. The date of COVID-19 occurring is March 11, 2020. 9. The overall period is from March 26, 2013, to March 11, 2021.

Moreover, all commodities have negative (respectively, positive) values of 'Mean' or 'R$^a$' in the pre-pandemic (respectively, post-pandemic) period. These results imply the following two implications. Firstly, the value of mean return in the overall period is approximately the average values of mean return in the two subperiods, and so are risk-adjusted return and standard deviation. Secondly, during the post-pandemic period, this pandemic increased the assets' risk measured by the standard deviation, and the Quantitative Easing (QE) executed after this pandemic crisis raised the return and risk-adjusted return of assets.

Regarding the second type of performance comparison, we discover the following two phenomena. Firstly, regarding the overall period, soybean oil (respectively, corn) owns the largest (respectively, lowest) values of 'Mean' or 'R$^a$' among the seven assets whereas coffee (respectively, soybean) owns the largest (respectively, lowest) values of 'SD'. Notably, only soybean oil has a positive value of 'Mean' and 'R$^a$'. The reason may be that recently soybean oil has played

a prominent role in alternative fuels. Secondly, among seven assets, coffee (respectively, soybean oil) owns the greatest (respectively, smallest) values of 'R$^a$' for the pre-pandemic period whereas coffee (respectively, soybean oil) is changed into have the smallest (respectively, greatest) values of 'R$^a$' for the post-pandemic period. These phenomena imply the following implication. Firstly, regarding the overall period, among all commodities soybean oil (respectively, coffee) owns the largest values of risk-adjusted return (respectively, standard deviation). Secondly, the two commodities, soybean oil and coffee, are easily influenced by this pandemic. Because, among seven assets, coffee owns the largest value of risk-adjusted return in the pre-pandemic period whereas it is changed into possesses the smallest value of risk-adjusted return in the post-pandemic period. The same phenomenon also occurs in soybean oil. From the discussion of the results of the above two types of performance comparison, this pandemic impacts the investment attributes of seven commodities in the agriculture market. The above results confirm that according to the date of the COVID-19 pandemic occurring, it is reasonable to partition the study period into the pre-pandemic and post-pandemic periods.

About the remaining descriptive statistics, the following phenomena are found. The return's distribution exhibits right- or left-skewed and bears a greater and thicker tail than the normal distribution as shown by the coefficients of skewness and excess kurtosis. This indicates that the distribution of the return series doesn't follow a normal distribution, and the J-B normality test statistics substantiate this result [33]. Additionally, as reported by the Ljung-Box $Q^2$ (24) statistics of the squared returns, the return series displays linear dependence and a strong ARCH effect. From the above discoveries, they nearly own the identical characteristics as those for most of the financial return series and a GARCH family model is favorable to capturing the time-varying volatility and fat tails discovered in these asset return series [27–30]. This indicates that it is acceptable to use the bivariate GARCH model in this study.

Fig 1 depicts the trend of both returns and price levels for seven commodities across the overall period. From Fig 1, we find that, because of the QE executed after the COVID-19 pandemic, the price of commodities experienced a fast rise after this pandemic. We also discover that the volatility clustering appears significantly across the overall period. The phenomena found above are almost consistent with those obtained in Table 1. Regarding 21 pairs of assets in the agricultural market, Fig 2 depicts the trend of price levels of two commodities during the overall period. In Fig 2, we find that, regarding each of the 21 pairs of assets, the price levels of two commodities have nearly the same trend across the overall period, indicating that the interactive relationship between any two commodities within the agricultural market is closely related. The 21 paired commodities are composed of any two assets among seven commodities in the agricultural market. This implies that it is meaningful to investigate the interactions among the commodities in the agriculture market.

## 4. Empirical results

In this section, we utilize the bivariate GARCH model's results for 21 pairs of data during two subperiods to explore the following two questions for the pre-pandemic and post-pandemic periods. Among seven commodities, which commodity is the net transmitter or net recipient on the return spillover (or volatility spillover) network, and which commodity is the most or least active in the networks of volatility and return spillovers as well as correlation? The above results are used to debate the influence of the COVID-19 pandemic on the interactions between the commodities in the agricultural market from the viewpoints of net transmitters, net receivers, and activators in a connectedness network. To complete the above job, we transfer the significant situations of values of parameters connected with three types of interactions into the corresponding symbolic results. (Notably, as mentioned in section2: Methodology,

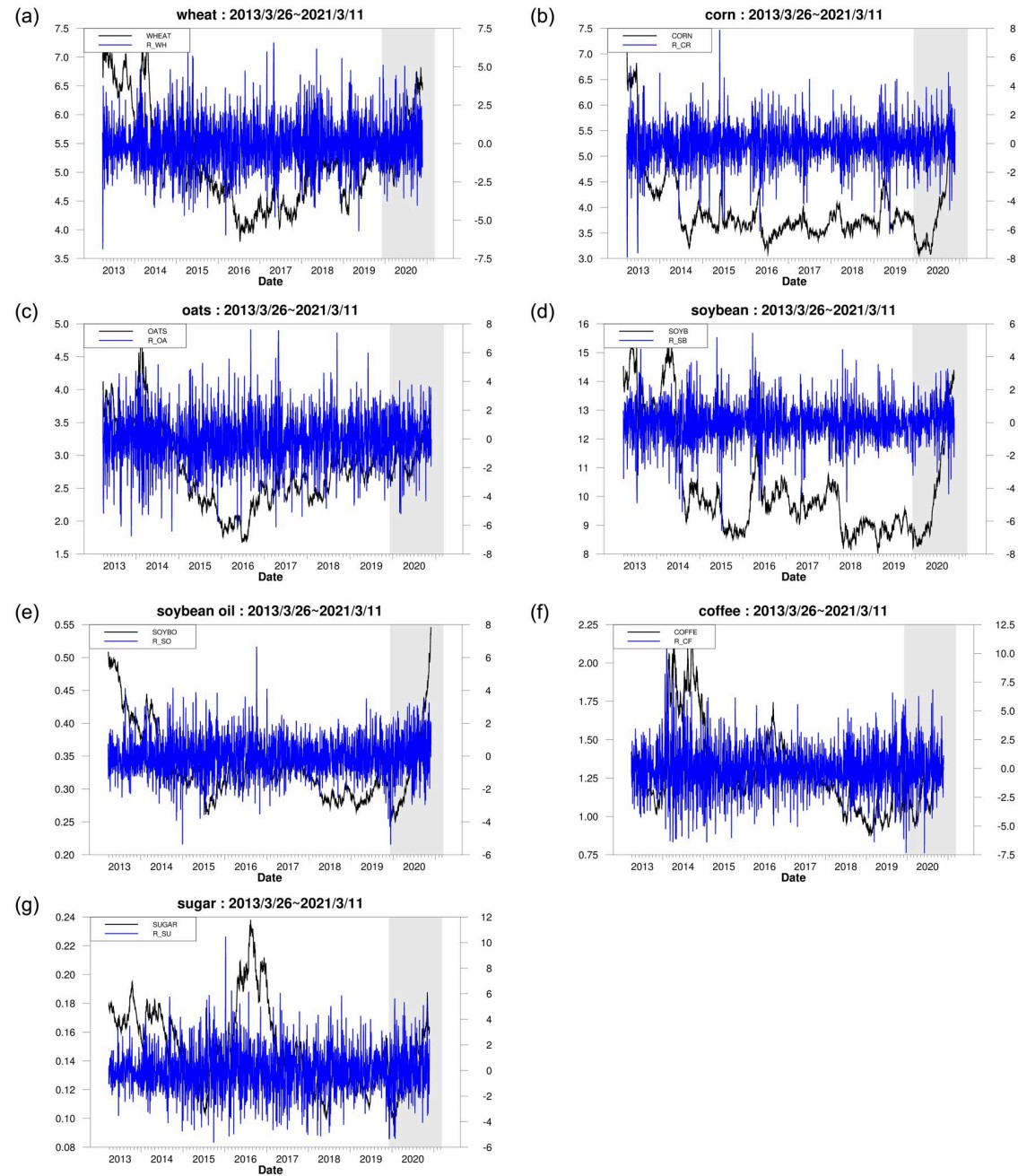

**Fig 1. The trend of price level and return of agriculture commodities.**

parameters connected with the return spillover are '$\phi_{12}^{B}$ and $\phi_{21}^{B}$' for the pre-pandemic period but '$\phi_{12}^{A}$ and $\phi_{21}^{A}$' for the post-pandemic period. Hence, we use the significant situations of both values of parameters '$\phi_{12}^{B}$ and $\phi_{21}^{B}$' to investigate the state of return spillover in the pre-pandemic period and so are parameters '$\phi_{12}^{A}$ and $\phi_{21}^{A}$' for the post-pandemic period. The same process is also applied for the parameters related to volatility spillover and correlation.). Then, we follow [10, 11, 22, 25] to plot six network diagrams of volatility spillover, return spillover, and

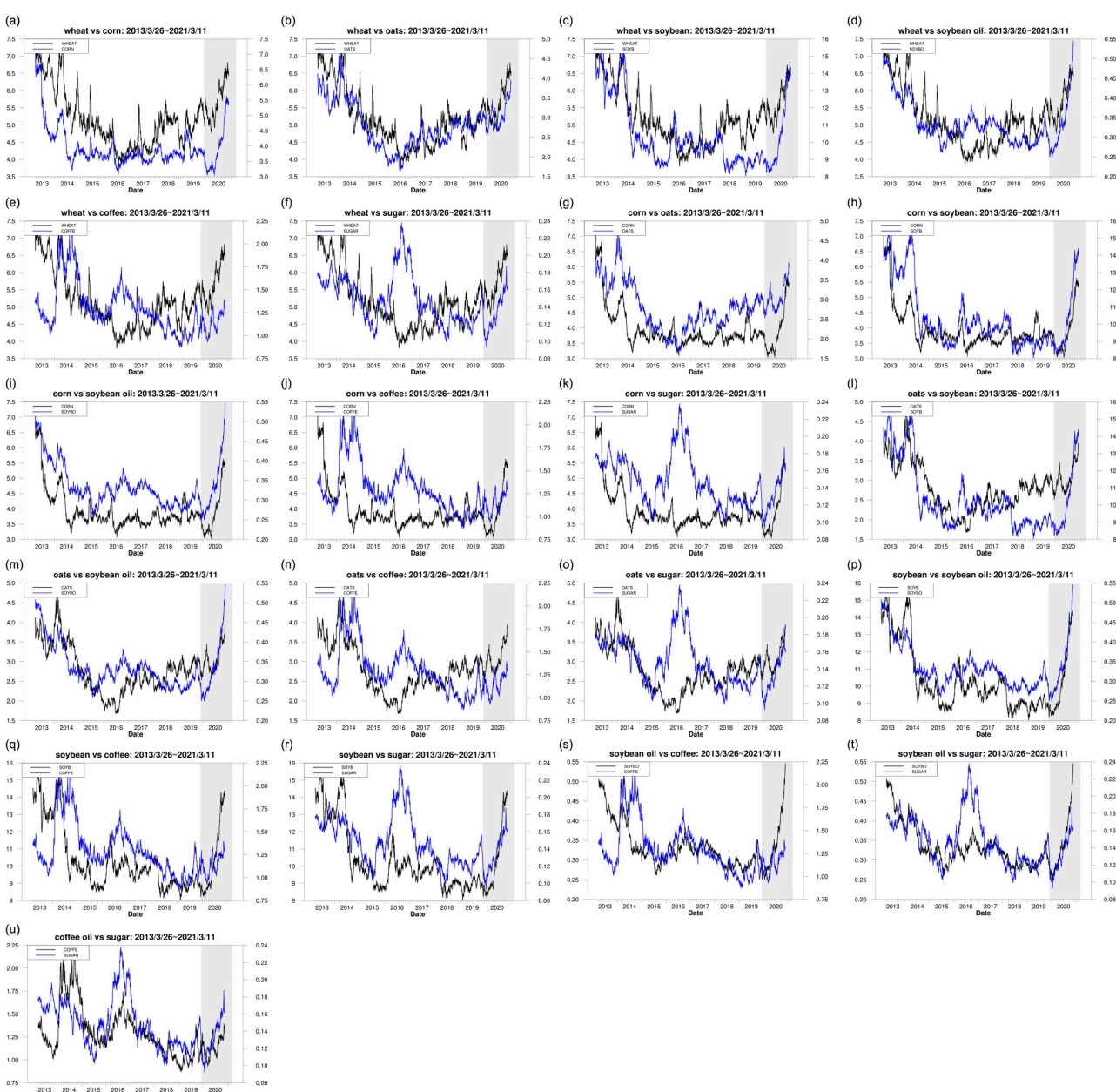

**Fig 2. The trend of price levels for a pair of agriculture commodities.**

correlation for the pre-pandemic and post-pandemic periods by using the above symbolic results. The above process is illustrated in the subsequent subsections.

## 4.1 The influence of the COVID-19 pandemic on the return connectedness

In this subsection, we take some examples in Table 2 to illustrate how to transfer the significant situations of values of parameters related to return spillover into the symbolic results. Then, we use the above symbolic results for the pre-pandemic and post-pandemic periods to plot two return spillover network diagrams as illustrated in Fig 3(a) and 3(b), respectively.

**Table 2. The summary results of return spillover for the 21 pairs of assets.**

| | wh-cr | wh-oa | wh-sb | wh-so | wh-cf | wh-su | cr-oa |
|---|---|---|---|---|---|---|---|
| $\phi_{12}^{B}$ | -0.0076 (0.026) | -0.0210 (0.020) | **-0.0186 (0.030)** | 0.0201 (0.030) | **0.0137 (0.017)** | 0.0023 (0.019) | 0.0128 (0.015) |
| $\phi_{21}^{B}$ | 0.0040 (0.019) | **-0.0629 (0.026)[b]** | 0.0104 (0.016) | 0.0139 (0.016) | -0.0000 (0.025) | 0.0182 (0.023) | -0.0402 (0.029) |
| pre | × | →<br>(−) | × | × | × | × | × |
| $\phi_{12}^{A}$ | **-0.1341 (0.067)[b]** | **-0.0516 (0.059)** | 0.0095 (0.087) | **-0.1340 (0.072)[a]** | 0.0039 (0.043) | **0.0539 (0.055)** | **0.0183 (0.046)** |
| $\phi_{21}^{A}$ | **0.0572 (0.048)** | 0.0303 (0.057) | **0.0611 (0.036)[a]** | **0.0219 (0.053)** | **0.1540 (0.066)[b]** | **0.0320 (0.070)** | **-0.1879 (0.059)[c]** |
| post | ←<br>(−) | × | →<br>(+) | ←<br>(−) | →<br>(+) | × | →<br>(−) |
| | cr-sb | cr-so | cr-cf | cr-su | oa-sb | oa-so | oa-cf |
| $\phi_{12}^{B}$ | 0.0045 (0.030) | -0.0013 (0.026) | 0.0293 (0.014)[b] | 0.0008 (0.016) | -0.0328 (0.035) | -0.0125 (0.034) | **0.0384 (0.022)[a]** |
| $\phi_{21}^{B}$ | -0.0072 (0.021) | 0.0052 (0.019) | -0.0098 (0.029) | 0.0283 (0.027) | 0.0123 (0.014) | 0.0005 (0.014) | 0.0087 (0.024) |
| pre | × | × | ←<br>(+) | × | × | × | ←<br>(+) |
| $\phi_{12}^{A}$ | **0.0676 (0.063)** | **0.0128 (0.047)** | **0.0677 (0.045)** | **0.0860 (0.044)[a]** | **-0.1579 (0.061)[b]** | **-0.1927 (0.066)[c]** | -0.0150 (0.045) |
| $\phi_{21}^{A}$ | **0.0474 (0.040)** | **-0.0123 (0.060)** | **0.0313 (0.085)** | **0.0903 (0.088)** | **-0.0285 (0.033)** | **0.0050 (0.049)** | **0.0981 (0.078)** |
| post | × | × | × | ←<br>(+) | ←<br>(−) | ←<br>(−) | × |
| | oa-su | sb-so | sb-cf | sb-su | so-cf | so-su | cf-su |
| $\phi_{12}^{B}$ | 0.0271 (0.025) | **-0.0083 (0.020)** | 0.0020 (0.012) | 0.0129 (0.014) | 0.0242 (0.012)[a] | **0.0053 (0.014)** | 0.0211 (0.022) |
| $\phi_{21}^{B}$ | **-0.0325 (0.020)** | 0.0047 (0.021) | 0.0374 (0.034) | **0.0441 (0.032)** | 0.0715 (0.037)[a] | 0.0578 (0.036) | **0.0116 (0.017)** |
| pre | × | × | × | × | ↔<br>(+) | × | × |
| $\phi_{12}^{A}$ | **0.0607 (0.047)** | 0.0049 (0.032) | **0.0605 (0.031)[a]** | **0.0285 (0.027)** | **0.0507 (0.044)** | 0.0050 (0.041) | **0.0285 (0.068)** |
| $\phi_{21}^{A}$ | -0.0229 (0.070) | **-0.0273 (0.062)** | **0.0642 (0.107)** | -0.0044 (0.105) | **0.1048 (0.086)** | **0.0796 (0.079)** | 0.0115 (0.049) |
| post | × | × | ←<br>(+) | × | × | × | × |

*Notes*: 1. The 'wh', 'cr', 'oa', 'sb', 'so', 'cf', and 'su' denote the wheat, corn, oat, soybean, soybean oil, coffee, and sugar in the agricultural market, respectively. 2. The superscripts a, b, and c on a parameter estimate denote the parameter estimate is significantly at the 10%, 5%, and 1% levels, respectively. Numbers in parentheses at the row of parameters are standard errors. 3. The superscripts 'B' and 'A' on a parameter denote that the parameter corresponds to the pre-pandemic and post-pandemic periods, respectively. 4. The symbol '×' represents that the interaction (correlation and return and volatility spillovers) of a pair of assets does not exist. 5. The symbol '→' in row 'pre' (respectively, 'post') denotes that the return spillover from the first asset to the second asset significantly exists for a pair of assets during the pre-pandemic (respectively, post-pandemic) period if the value of parameter '$\phi_{21}^{B}$ (respectively, '$\phi_{21}^{A}$') is significant. 6. The symbol '←' in rows 'pre' (respectively, 'post') denotes that the return spillover from the second asset to the first asset significantly exists for a pair of assets during the pre-pandemic (respectively, post-pandemic) period if the value of parameter '$\phi_{12}^{B}$ (respectively, '$\phi_{12}^{A}$') is significant. 7. The symbol '+' (respectively, '-') inside the bracket underneath the symbol '→' or '←' in rows 'pre' and 'post' denotes that the return spillover is significantly positive (respectively, negative). 8. Bold font marked in estimates denotes that this estimate owns the greater value of return spillover in absolute value when the values of parameters $\phi_{12}^{B}$ and $\phi_{12}^{A}$ respectively denoting the return spillover for the pre-pandemic and post-pandemic periods are compared with each other so are the case for parameters $\phi_{21}^{B}$ and $\phi_{21}^{A}$.

Regarding 21 pairs of data, Table 2 lists the values of parameters '$\phi_{12}^{B}$ and $\phi_{21}^{B}$' and '$\phi_{12}^{A}$ and $\phi_{21}^{A}$' of the bivariate GARCH model and the corresponding symbolic results. For example, regarding the 'wh-oa' pair of data in Table 2, only the value of parameter $\phi_{21}^{B}$ (-0.0629) is negative significantly. This result indicates that a negative return spillover from the first asset, wheat, to the second asset, oat, exists in the pre-pandemic period. Then, in Table 2 we record the symbol '→' in the column 'wh-oa' and the row 'pre' and we use a blue arrow from the red node (−) 'Wheat' to the yellow node 'Oat' in Fig 3(a) to represent this negative spillover. Moreover, regarding the cr-cf pair of data, only the value of parameter $\phi_{12}^{B}$ (0.0293) is significantly positive. This result indicates that a positive return spillover from the second asset, coffee, to the first asset, corn, exists in the pre-pandemic period. Then, in Table 2 we record the symbol '←' (+)

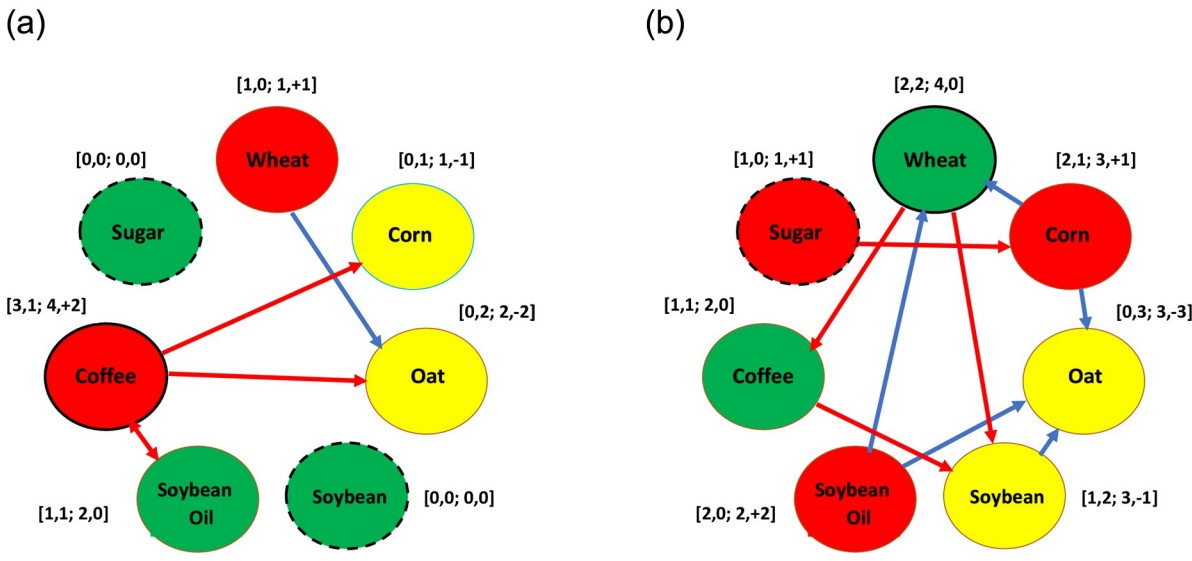

**Fig 3.** The return spillover network (a) Pre-pandemic period. (b) Post-pandemic period.

in the column 'cr-cf' and the row 'pre' and we utilize a red arrow from the red node 'Coffee' to the yellow node 'Corn' in Fig 3(a) to denote this positive spillover.

Subsequently, regarding each node in Fig 3(a) and 3(b), we calculate the total number of arrows, that are far away from that node (respectively, point into that node), then we record this number in the first (respectively, second) position in the bracket beside this node. Finally, we do the summation and difference for the first and second numbers in the bracket, and then we record these two results in the third and fourth positions in the same bracket, respectively. For example, regarding the red node 'Coffee' in Fig 3(a), we find there exist three arrows from node 'Coffee' to nodes 'Corn', 'Oat', and 'Soybean Oil', which are all far away from node 'Coffee'. On the other hand, there also exists an arrow from node 'Soybean Oil' to node 'Coffee', which points to node 'Coffee'. Then, we record the numbers '3' and '1' respectively in the first and second positions of the bracket beside the red node 'Coffee'. Hence, the summation and difference for the first and second numbers, '3' and '1', are 4 and +2, respectively. Then, we record the numbers '4' and '+2' respectively in the third and fourth positions of the bracket beside the red node 'Coffee'.

From Fig 3(a), we get the following results. **Firstly**, the third number in the bracket beside the red node 'Coffee', 4, is the greatest among the third numbers for seven nodes. This indicates that, among the seven commodities, coffee is the most closely related to the other commodities from the viewpoint of coffee affecting the other commodities and coffee being affected by the other commodities. Hence, we regard coffee as an activator in this return spillover network diagram. **Secondly**, the third number in the brackets beside the green nodes 'Sugar' and 'Soybean', 0, is the smallest among the third numbers for seven nodes. This indicates that, among the seven commodities, both sugar and soybean are the least closely related to the other commodities. Thus, in the pre-pandemic period coffee (respectively, sugar and soybean) is the most (respectively, least) active asset on the return spillover within seven commodities. If a node represents the most (least) active asset in this network, then the outer circle of this node is marked in a solid (dash) line. **Thirdly**, the fourth numbers in the brackets beside red nodes 'Wheat' and 'Coffee' are equal to +1 and +2, respectively, which are all greater than zero. This result indicates that wheat or coffee can affect the other commodities more than it is

affected by the other commodities. Hence, we regard wheat and coffee as the net transmitters in the return spillover network diagram in the pre-pandemic period. **Fourthly**, the fourth numbers in the brackets beside the yellow nodes 'Corn' and 'Oat' are equal to -1 and -2, respectively, which are all less than zero. This indicates that the other commodities can affect corn (or oat) more than corn (or oat) affect the other commodities. Thus, we regard both corn and oat as the net recipients in the return spillover network diagram in the pre-pandemic period. **Finally**, the fourth numbers in the brackets beside the green nodes 'Sugar', 'Soybean', and 'Soybean Oil' are equal to zero. Thus, we regard them as neutrals in the return spillover network diagram because they are neither net transmitters nor net recipients.

Using the alike inference procedure in Fig 3(a), the following results are obtained from Fig 3(b). **Firstly**, the third numbers in the brackets beside the green node 'Wheat' and the red node 'Sugar' respectively equal 4 and 1, the greatest and smallest numbers among the third numbers for seven nodes. This indicates that, in the post-pandemic period, wheat (sugar) is the most (least) active asset within the seven commodities on the return spillover network. Thus, we regard wheat as an activator in this return spillover network. **Secondly**, the fourth numbers in the brackets beside the red nodes 'Corn', 'Soybean Oil', and 'Sugar' respectively are equal to +1, +2, and +1, which are all greater than zero. On the other hand, the fourth numbers in the brackets beside the yellow nodes 'Oat' and 'Soybean' respectively are equal to -3 and -1, which are all less than zero. In addition, the fourth numbers in the brackets beside the green nodes 'Wheat' and 'Coffee' are all equal to 0. The above results indicate that for the return spillover network in the post-pandemic period, oat and soybean are the net recipients whereas corn, soybean oil, and sugar are the net transmitters. In addition, wheat and coffee are neutrals.

From the above discussion, the activators (or net transmitters) in the return spillover network for the two subperiods are completely different. The reason are the most active commodity in the pre-pandemic period was coffee but wheat for the post-pandemic period. Conversely, the net transmitters of return spillover in the pre-pandemic period were wheat and coffee but corn, soybean oil, and sugar for the post-pandemic period. This indicates that Hypothesis 1 and Hypothesis 2 are not rejected in the return spillover network. On the other hand, the net recipients in the return spillover network for the two subperiods are almost different. The reasons are that the net recipients of return spillover in the pre-pandemic period were corn and oat but oat and soybean for the post-pandemic period. Notably, oat was concurrently the net recipient of return spillover for these two periods. This indicates that Hypothesis 3 is almost not rejected in the return spillover network. The above results imply that the COVID-19 pandemic impacts the return connectedness of commodities in the agriculture market.

In addition, from Fig 3(a) and 3(b), the following results are found. **Firstly**, corn is the net receiver in the pre-pandemic period but the net transmitter in the post-pandemic period, indicating that the COVID-19 pandemic alters the commodities' role in the return connectedness. **Secondly**, the total number of arrows in Fig 3(b) is greater than that in Fig 3(a), especially that for the blue arrows, indicating that this pandemic boosts the interactive degree of the unidirectional spillovers, especially for negative return spillover. Moreover, in Table 2, we find that the values of parameters $\phi_{12}^A$ and $\phi_{21}^A$ are respectively greater than those of parameters $\phi_{12}^B$ and $\phi_{21}^B$ in the absolute value for most cases. This implies that this pandemic enhances the intensity of return spillover. This result is in harmony with that found by [10–12, 24, 25, 34] but is similar to that found by [35]. Because most of them found that these spillovers are always intensified in volatile periods caused by economic and political events such as the GFC in [12, 24, 34] as well as the COVID-19 pandemic in [11, 24]. Notably, sugar is the most suitable hedged asset

on the return of the other commodities in the agriculture market because sugar is the least active asset within seven commodities on the return spillover and thus it is weakly connected with the other commodities in the pre-pandemic and post-pandemic periods.

## 4.2 The influence of the COVID-19 pandemic on the volatility connectedness

In this subsection, 'the process of transferring the significant situations of values of parameters related to volatility spillover into the corresponding symbolic results' and 'the theory of plotting a volatility spillover network diagram' are the same as those for the case of return spillover mentioned in the subsection 4.1: The Influence of the COVID-19 pandemic on the return connectedness. Hence, regarding 21 pairs of data, Table 3 lists the values of parameters '$v_{12}^B$ and $v_{21}^B$' and '$v_{12}^A$ and $v_{21}^A$' of the bivariate GARCH model and the corresponding symbolic results. Fig 4(a) and 4(b) are the volatility spillover network diagrams for the pre-pandemic and post-pandemic periods, respectively. Accordingly, in this subsection, we only take one example that

**Table 3. The summary results of volatility spillover for the 21 pairs of assets.**

| | wh-cr | wh-oa | wh-sb | wh-so | wh-cf | wh-su | cr-oa |
|---|---|---|---|---|---|---|---|
| $v_{12}^B$ | **-0.0269** (0.008)$^c$ | 0.0193 (0.006)$^c$ | **-0.0122** (0.007) | 0.0134 (0.013) | 0.0041 (0.004) | 0.0115 (0.009) | 0.0142 (0.009) |
| $v_{21}^B$ | 0.0036 (0.006) | -0.0188 (0.031) | **0.0129** (0.002)$^c$ | 0.0002 (0.002) | -0.0050 (0.006) | 0.0093 (0.007) | -0.0044 (0.021) |
| pre | ←(−) | ←(+) | →(+) | × | × | × | × |
| $v_{12}^A$ | -0.0092 (0.014) | **0.0349** (0.013)$^b$ | -0.0039 (0.017) | **0.0212** (0.013) | **0.0087** (0.005) | **0.0127** (0.009) | **0.0264** (0.014)$^a$ |
| $v_{21}^A$ | **0.0132** (0.008) | **-0.0700** (0.034)$^b$ | 0.0118 (0.003)$^c$ | **0.0041** (0.004) | **-0.0062** (0.008) | **0.0156** (0.011) | **-0.0663** (0.025)$^c$ |
| post | × | ←→(+)(−) | →(+) | × | × | × | ←→(+)(−) |
| | cr-sb | cr-so | cr-cf | cr-su | oa-sb | oa-so | oa-cf |
| $v_{12}^B$ | 0.6805 (0.030)$^c$ | 0.0078 (0.016) | 0.0033 (0.002) | **-0.0143** (0.008) | 0.0173 (0.028) | -0.0256 (0.108) | **0.0869** (0.028)$^c$ |
| $v_{21}^B$ | **0.1237** (0.017)$^c$ | 0.0017 (0.002) | -0.0006 (0.008) | **-0.0262** (0.010)$^b$ | 0.0201 (0.008)$^b$ | 0.0032 (0.003) | 0.0586 (0.031)$^a$ |
| pre | ↔(+) | × | × | →(−) | →(+) | × | ↔(+) |
| $v_{12}^A$ | **0.7379** (0.084)$^c$ | **0.0138** (0.013) | **0.0057** (0.003) | -0.0057 (0.007) | **-0.0527** (0.021)$^b$ | **-0.1336** (0.080)$^a$ | 0.0120 (0.030) |
| $v_{21}^A$ | 0.1114 (0.020)$^c$ | **0.0065** (0.004) | **0.0007** (0.011) | -0.0024 (0.013) | **0.0305** (0.011)$^c$ | **0.0074** (0.004)$^a$ | **0.0790** (0.041)$^a$ |
| post | ↔(+) | × | × | × | ←→(−)(+) | ←→(−)(+) | →(+) |
| | oa-su | sb-so | sb-cf | sb-su | so-cf | so-su | cf-su |
| $v_{12}^B$ | **0.0320** (0.038) | **-0.0084** (0.001)$^c$ | **0.0019** (0.000)$^c$ | -0.0008 (0.002) | 0.0031 (0.000)$^c$ | **-0.0014** (0.001) | **-0.0122** (0.001)$^c$ |
| $v_{21}^B$ | -0.0002 (0.012) | 0.0018 (0.000)$^b$ | **-0.0108** (0.008) | -0.0038 (0.007) | **0.0055** (0.005) | 0.0101 (0.006) | -0.0017 (0.002) |
| pre | × | ←→(−)(+) | ←(+) | × | ←(+) | × | ←(−) |
| $v_{12}^A$ | -0.0159 (0.031) | -0.0069 (0.001)$^c$ | 0.0007 (0.001) | **-0.0025** (0.002) | **0.0044** (0.001)$^c$ | -0.0005 (0.002) | -0.0104 (0.004)$^c$ |
| $v_{21}^A$ | **0.0071** (0.017) | **0.0086** (0.005) | -0.0103 (0.014) | **0.0210** (0.021) | 0.0043 (0.008) | **0.0183** (0.010)$^a$ | **0.0050** (0.005) |
| post | × | ←(−) | × | × | ←(+) | →(+) | ←(−) |

*Notes*: 1 See the notes 1–4 of Table 2. 2. The symbol '→' in row 'pre' (respectively, 'post') denotes that the volatility spillover from the first asset to the second asset significantly exists for a pair of assets during the pre-pandemic (respectively, post-pandemic) period if the value of parameter '$v_{21}^B$ (respectively, '$v_{21}^A$') is significant. 3. The symbol '←' in rows 'pre' (respectively, 'post') denotes that the volatility spillover from the second asset to the first asset significantly exists for a pair of assets during the pre-pandemic (respectively, post-pandemic) period if the value of parameter '$v_{12}^B$' (respectively, '$v_{12}^A$') is significant. 4. The symbol '+' (respectively, '-') inside the bracket underneath the symbol '→' or '←' in rows 'pre' and 'post' denotes that the volatility spillover is significantly positive (respectively, negative). 5. Bold font marked in estimates denotes that this estimate owns the greater value of volatility spillover in absolute value when the values of parameters $v_{12}^B$ and $v_{12}^A$ respectively denoting the volatility spillover for the pre-pandemic and post-pandemic periods are compared with each other so are the case for parameters $v_{21}^B$ and $v_{21}^A$.

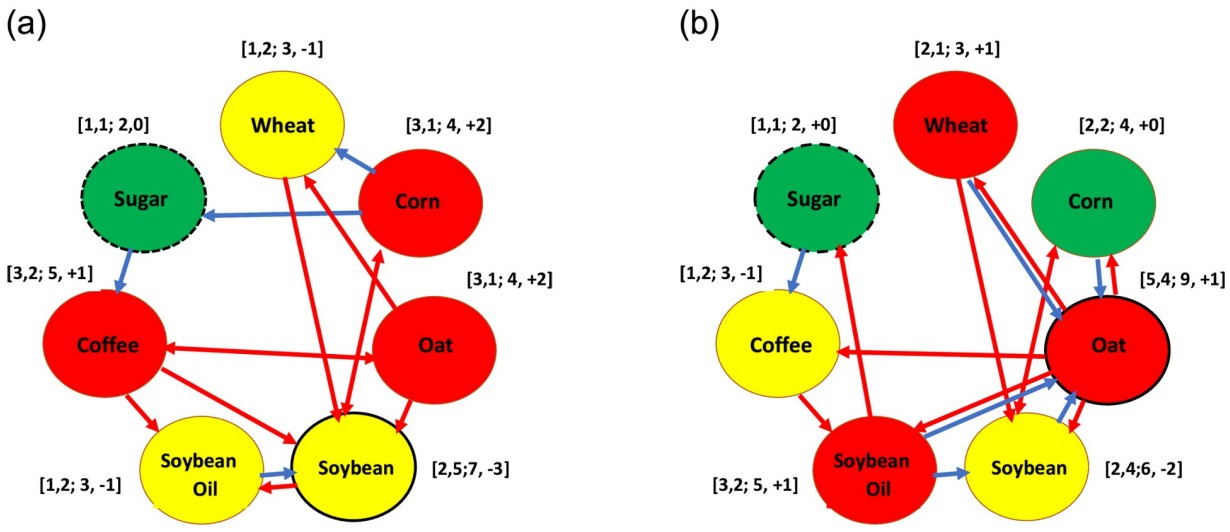

**Fig 4.** The volatility spillover network (a) Pre-pandemic period. (b) Post-pandemic period.

doesn't appear in subsection 4.1: The Influence of the COVID-19 pandemic on the return connectedness to illustrate. It is the bidirectional volatility spillovers, which have different signs (positive or negative) for different directional spillovers. For example, regarding the 'sb-so' pair of data, the value of parameter $v_{12}^{B}$ (-0.0084) is negative significantly but the value of parameter $v_{21}^{B}$ (0.0018) is significantly positive. This implies that, in the pre-pandemic period, a negative volatility spillover subsists from the second asset, soybean oil, to the first asset, soybean, and a positive volatility spillover exists from the first asset, soybean, to the second asset, soybean oil. Then, in Table 3 the symbols '$\underset{(-)}{\leftarrow}$' and '$\underset{(+)}{\rightarrow}$' are recorded in the row 'pre' and the column 'sb-so'. Subsequently, in Fig 4(a) the above symbol '$\underset{(-)}{\leftarrow}$' is represented by a blue arrow from the yellow node 'Soybean Oil' to the yellow node 'Soybean' whereas another symbol '$\underset{(+)}{\rightarrow}$' is depicted by a red arrow from the yellow node 'Soybean' to the yellow node 'Soybean Oil'. Finally, I perform four types of calculation for each node in Fig 4(a) and 4(b) by imitating the same process in Fig 3(a) and 3(b). The results of four types of calculation are listed in the bracket beside each node.

From Fig 4(a), we get the following results. **Firstly**, the third numbers in the brackets beside the yellow node 'Soybean' and the green node 'Sugar' respectively equal 7 and 2, the greatest and smallest numbers among the third numbers for seven nodes. This indicates that, in the pre-pandemic period, soybean (respectively, sugar) is the most (respectively, least) active asset within seven commodities on the volatility spillover network. Moreover, we regard soybean as an activator in this volatility spillover network. **Secondly**, the fourth numbers in the brackets beside the red nodes 'Corn', 'Oat', and 'Coffee' respectively are equal to +2, +2, and +1, which are all greater than zero. On the other hand, the fourth numbers in the brackets beside the yellow nodes 'Wheat', 'Soybean', and 'Soybean Oil' respectively are equal to -1, -3, and -1, which are all less than zero. In addition, the fourth number in the bracket beside the green node 'Sugar' is equal to 0. These results indicate that regarding the volatility spillover network in the pre-pandemic period, corn, oat, and coffee are the net transmitters but soybean, soybean oil, and wheat are the net recipients. In addition, sugar is neutral.

From Fig 4(b), we get the following results. **Firstly**, the third numbers in the brackets beside the red node 'Oat' and the green node 'Sugar' respectively equal 9 and 2, the greatest and smallest numbers among the third numbers for seven nodes. This indicates that, in the post-pandemic period, oat (respectively, sugar) is the most (respectively, least) active asset within seven commodities on the volatility spillover network. Moreover, we regard oat as an activator in this volatility spillover network. **Secondly**, the fourth numbers in the brackets beside the red nodes 'Wheat', 'Oat', and 'Soybean Oil' respectively are equal to +1, +1, and +1, which are all greater than zero. On the contrary, the fourth numbers in the brackets beside the yellow nodes 'Soybean' and 'Coffee' respectively are equal to -2 and -1, which are all less than zero. In addition, the fourth numbers in the brackets beside the green nodes 'Sugar' and 'Corn' are all equal to 0. These results indicate that regarding the volatility spillover network in the post-pandemic period, wheat, oat, and soybean oil are the net transmitters whereas soybean and coffee are the net recipients. Additionally, sugar and corn are neutrals.

To sum up, the activators in the volatility spillover networks for the two subperiods are different because the activators in the pre-pandemic period are soybean but oat in the post-pandemic period. This result implies that Hypothesis 4 isn't rejected in the volatility spillover network. Moreover, corn, oat, and coffee are the net transmitters of volatility spillover in the pre-pandemic period but wheat, oat, and soybean oil in the post-pandemic period. Only oat simultaneously appears in these two subperiods. On the contrary, soybean, wheat, and soybean oil are the net recipients of volatility spillover in the pre-pandemic period but soybean and coffee in the post-pandemic period. Notably, soybean simultaneously appears in these two subperiods. That is, irrespective of the net transmitters or net recipients, they are almost different in the two subperiods. These results indicate that Hypothesis 5 and Hypothesis 6 are almost not rejected in the volatility spillover network. The above results infer that the COVID-19 pandemic impacts the volatility connectedness of commodities in the agriculture market.

In addition, from Fig 4(a) and 4(b), we find the following phenomena. Firstly, both wheat and soybean oil are the net receivers in the pre-pandemic period but the net transmitters in the post-pandemic period. Moreover, coffee is the net transmitter in the pre-pandemic period but the net receiver in the post-pandemic period. These results indicate that the COVID-19 pandemic changes the commodities' roles in the volatility connectedness. Secondly, the total number of two-way arrows in Fig 4(b) is greater than that in Fig 4(a) and the two-way arrows have different colors in different directions. This implies that this pandemic enhances the interactive degree of the bidirectional volatility spillovers with positive and negative values in different directions. In addition, in Table 3, we find that the values of parameters $v_{12}^A$ and $v_{21}^A$ are respectively greater than those of parameters $v_{12}^B$ and $v_{21}^B$ in absolute value for most cases. This implies that this pandemic increases the intensity of volatility spillover. This result is in harmony with that found by [10–12, 24, 25] because they found that these spillovers are always intensified in volatile periods caused by economic and political events such as the GFC in [12, 24] as well as the COVID-19 pandemic in [24, 25]. Notably, sugar is the most suitable hedged asset on the risk of the other commodities in the agriculture market because sugar is weakly connected with the other commodities in the pre-pandemic and post-pandemic periods.

## 4.3 The influence of the COVID-19 pandemic on the correlation connectedness

In this subsection, we take one example in Table 4 to illustrate how to transfer the significant situations of values of parameters related to correlation into the symbolic results. Then, we use the above symbolic results for the pre-pandemic and post-pandemic periods to plot two correlation network diagrams as illustrated in Fig 5(a) and 5(b). Regarding 21 pairs of data, Table 4

**Table 4. The summary results of correlation for the 21 pairs of assets.**

| | wh-cr | wh-oa | wh-sb | wh-so | wh-cf | wh-su | cr-oa |
|---|---|---|---|---|---|---|---|
| $\omega_{12}^{B}$ | 0.0552 (0.016)[c] | 0.0345 (0.014)[b] | 0.0196 (0.003)[c] | 0.0745 (0.056) | **0.0112** (0.031) | 0.0031 (0.000)[c] | 0.0503 (0.014)[c] |
| pre | + | + | + | × | × | + | + |
| $\omega_{12}^{A}$ | **0.0609** (0.019)[c] | **0.0382** (0.017)[b] | **0.0272** (0.008)[c] | **0.1412** (0.115) | 0.0084 (0.024) | **0.0081** (0.002)[c] | **0.0565** (0.019)[c] |
| post | + | + | + | × | × | + | + |
| | cr-sb | cr-so | cr-cf | cr-su | oa-sb | oa-so | oa-cf |
| $\omega_{12}^{B}$ | 0.0228 (0.006)[c] | 0.0161 (0.007)[b] | 0.0123 (0.009) | 0.3878 (0.075)[c] | 0.0347 (0.011)[c] | 0.0120 (0.007) | 0.6393 (0.144)[c] |
| pre | + | + | × | + | + | × | + |
| $\omega_{12}^{A}$ | **0.0252** (0.008)[c] | **0.0294** (0.014)[b] | **0.0258** (0.021) | **1.1855** (0.273)[c] | **0.0657** (0.023)[c] | **0.0195** (0.013) | **0.7390** (0.391)[a] |
| post | + | + | × | + | + | × | + |
| | oa-su | sb-so | sb-cf | sb-su | so-cf | so-su | cf-su |
| $\omega_{12}^{B}$ | 0.1157 (0.075) | 0.0158 (0.001)[c] | **0.0132** (0.009) | 0.3081 (0.080)[c] | **0.0022** (0.001) | 0.0279 (0.013)[b] | **0.0547** (0.017)[c] |
| pre | × | + | × | + | × | + | + |
| $\omega_{12}^{A}$ | **0.1579** (0.182) | **0.0201** (0.003)[c] | 0.0100 (0.011) | **0.7531** (0.202)[c] | 0.0020 (0.003) | **0.0860** (0.049)[a] | 0.0528 (0.028)[a] |
| post | × | + | × | + | × | + | + |

*Notes*: 1 See the notes 1–4 of Table 2. 2. The symbol '+' in row 'pre' (respectively, 'post') denotes that the correlation is positive significantly for a pair of assets during the pre-pandemic (respectively, post-pandemic) period if the value of parameter $\omega_{12}^{B}$ (respectively, $\omega_{12}^{A}$) is significantly positive. 3. Bold font marked in estimates denotes that this estimate owns the greater value of covariance when the values of parameters $\omega_{12}^{B}$ and $\omega_{12}^{A}$ respectively denoting the covariances for the pre-pandemic and post-pandemic periods are compared with each other.

lists the values of parameters '$\omega_{12}^{B}$' and '$\omega_{12}^{A}$' of the bivariate GARCH model and the corresponding symbolic results. For example, for most pairs of data in Table 4, the value of parameter '$\omega_{12}^{B}$' is significant and positive such as the 'wh-cr'. This implies that in the pre-pandemic period, there exists a positive correlation in the above pair of data. Then, in Table 4 we record the symbol '+' in the column 'wh-cr' and the row 'pre', and in Fig 5(a), we use one red line between the node 'Wheat' and node 'Corn' to represent this positive correlation. Interestingly,

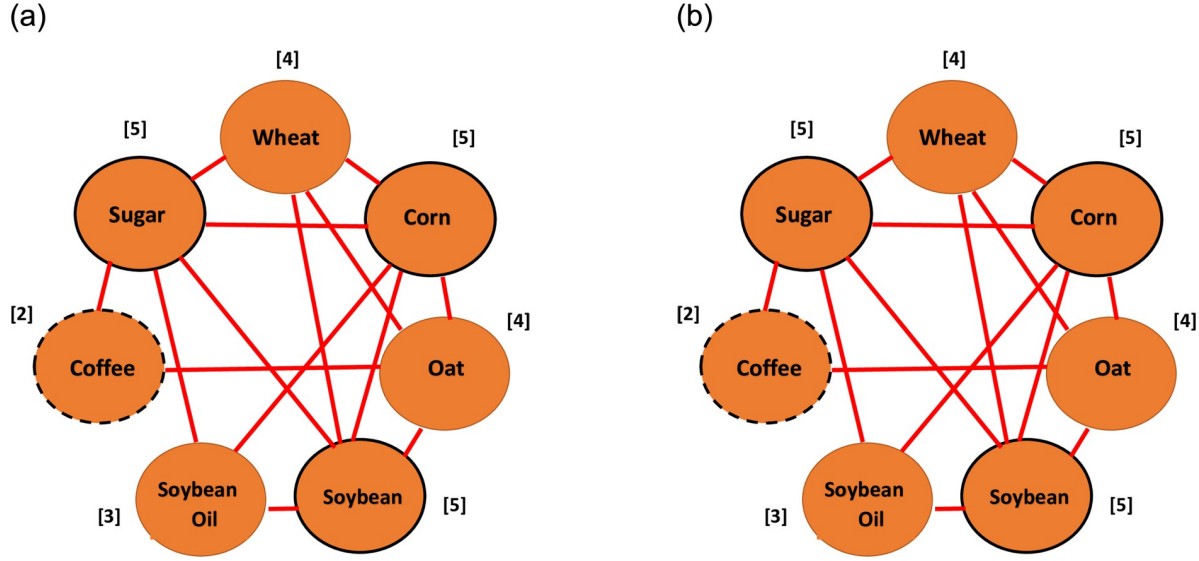

**Fig 5.** The correlation network (a) Pre-pandemic period. (b) Post-pandemic period.

we find the symbolic results in the rows 'pre' and 'post' are the same for each pair of data in Table 4. Hence, a correlation network diagram for the pre-pandemic period, Fig 5(a), is the same as that for the post-pandemic period, Fig 5(b). Subsequently, regarding each node in Fig 5(a) and 5(b), we calculate the total number of lines, which connect this node to the other nodes and record this number in the bracket beside this node. For example, regarding the node 'Corn' in Fig 5(a), we find there are five lines, which connect node 'Corn' to the other five nodes such as nodes 'Wheat', 'Sugar', 'Soybean Oil', 'Soybean', and 'Oat' and we record this number '5' in the bracket beside the node 'Corn'.

As shown in Fig 5(a) and 5(b), we find that the numbers in the brackets beside nodes 'Corn', 'Soybean', and 'Sugar' are all equal to 5, which is the greatest number among this number for seven nodes. This indicates that, in the two subperiods, corn, soybean, and sugar are the most active assets within seven commodities and we call them the activators in the correlation network. Conversely, we find that the number in the brackets beside node 'Coffee' is equal to 2, which is the smallest number among this number for seven nodes. This indicates that, in the two subperiods, coffee is the least active asset among the seven commodities on the correlation network.

To sum up, the most active commodities for the two subperiods are completely the same and they are corn, soybean, and sugar. The reasons are Fig 5(a) and 5(b) are completely the same and no pair of data is significantly changed by this pandemic. This indicates that Hypothesis 7 is rejected in the correlation network. This implies that this pandemic doesn't impact the correlation connectedness of commodities in the agriculture market. This phenomenon infers that short-term extreme events like the COVID-19 pandemic can't affect one long-term interaction such as correlation. However, in Table 4, we find that the values of parameters $\omega_{12}^B$ and $\omega_{12}^A$ are all positive and the value of parameters $\omega_{12}^A$ is greater than that of parameters $\omega_{12}^B$ for most cases. This indicates that the commodities in the agriculture market have positively interactive relationships and this pandemic enlarges the intensity of correlation. This result is in harmony with that found in [9, 15, 18] because they found the intensity of correlation increased in volatile periods caused by economic and political events such as the GFC [9, 15] and the COVID-19 pandemic [18]. Notably, coffee is the most suitable asset to achieve risk diversification for portfolios in the agricultural market because coffee is the least active asset in the correlation network for two subperiods.

## 5. Conclusion and further discussion

This study employs a bivariate GARCH model to inspect the Influence of the COVID-19 pandemic on the interactions between agricultural commodities from the viewpoint of net transmitters, net receivers, and activators in the connectedness networks. This study combines the advantages of methods used in the two categories of literature mentioned in section 1: Introduction to explore the short- and long-term interactions between two commodities. The empirical findings can be summarized as follows. Firstly, in the return spillover network, the activator in the pre-pandemic period is coffee but that in the post-pandemic period is wheat. The net transmitters in the pre-pandemic period are coffee and wheat but those in the post-pandemic period are soybean oil, corn, and sugar. The net receivers in the pre-pandemic period are oat and corn but those in the post-pandemic period are oat and soybean. Thus, Hypotheses 1–3 are not rejected in the return spillover network because the three major roles in the return connectedness network are almost different for the two subperiods. In addition, corn is the net receiver in the pre-pandemic period but the net transmitter in the post-pandemic period, indicating that the COVID-19 pandemic alters the roles of commodities in the return connectedness. Hence, the COVID-19 pandemic impacts the return connectedness of

commodities in the agriculture market. Notably, this pandemic also increases the interactive degree of unidirectional return spillovers, especially for negative return spillovers and it also increases the intensity of return spillovers.

Secondly, in the volatility spillover network, the activator in the pre-pandemic period is soybean but that in the post-pandemic period is oat. The net transmitters in the pre-pandemic period are corn, oats, and coffee but those in the post-pandemic period are wheat, oats, and soybean oil. The net receivers in the pre-pandemic period are soybean, wheat, and soybean oil but those in the post-pandemic period are soybean and coffee. Thus, Hypotheses 4–6 are not rejected in the volatility spillover network because the three major roles in the volatility connectedness network are almost different for the two subperiods. In addition, both wheat and soybean oil are the net receivers in the pre-pandemic period but the net transmitters in the post-pandemic period. Moreover, coffee is the net transmitter in the pre-pandemic period but the net receiver in the post-pandemic period. These results indicate that this pandemic changes the commodities' roles in the volatility connectedness. Hence, the COVID-19 pandemic impacts the volatility connectedness of commodities in the agriculture market. Notably, this pandemic increases the interactive degree of the bidirectional spillovers having positive and negative values in different directions. Moreover, it increases the intensity of volatility spillover.

Thirdly, in the correlation network, the activators for the pre-pandemic and post-pandemic periods are completely the same. They are corn, soybean, and sugar. Thus, Hypothesis 7 is rejected in the correlation network because the commodities of one major role in the correlation connectedness network are the same for the two subperiods. Hence, the COVID-19 pandemic doesn't impact the correlation connectedness of commodities in the agriculture market. In addition, the commodities in the agriculture market have positively interactive relationships and this pandemic increases the intensity of correlation.

In summary, in the agriculture market, the COVID-19 pandemic impacts the volatility and return connectedness of commodities but not the correlation connectedness of commodities. Thus, the COVID-19 pandemic, a short-term extreme event, can impact short-term interactions like volatility and return spillovers but not one long-term interaction like correlation. We can presume that the COVID-19 pandemic made a structural change in the volatility and return connectedness of commodities. This result can be supported by [36, 37]. For instance, Wu et al. found abnormal structural changes in the transmission mechanism within 36 national stock indexes during the COVID-19 crisis [36]. Moreover, regarding two subperiods, sugar is the least active asset in the volatility and return spillover networks, but coffee for the correlation network. The reasons are that 'sugar and coffee' and 'wheat, corn, oat, soybean, and soybean oil' are two groups that belong to the beverage and grain types of agricultural commodities, respectively. The two types of agricultural commodities own different attributes because 'Wheat, corn, oat, soybean, and soybean oil' are the main food products for human meals whereas 'sugar and coffee' are snack foods. Particularly, in the return spillover network, the soybean and soybean oil-the raw material of renewable diesel-are the neutrals in the pre-pandemic period, whereas the soybean and soybean oil are respectively changed into the net receiver and net transmitter in the post-pandemic period. Conversely, in the volatility spillover network, soybean and soybean oil are the net receivers in the pre-pandemic period whereas soybean oil is changed into the net transmitter in the post-pandemic period but doesn't change for soybean. This provides evidence that two raw materials of renewable diesel, especially for soybean oil increased their importance in the agricultural market during the post-pandemic period. Overall, the above impact results owing to this pandemic are obtained from the assumption of the variation of the average phenomena of the interactions in the pre-pandemic and post-pandemic periods. Therefore, the method in this study cannot observe how the

impact of this pandemic on the interactions varies with time. However, we can use the rolling window method with a given fixed sample window size to predict the parameters related to the interactions such as $\phi_{12}$, $\phi_{21}$, $\nu_{12}$, $\nu_{21}$, and $\omega_{12}$. From the trend of the values of the parameter estimate with time, we can observe how the impact of this pandemic on the interactions varies with time. This question can be reserved as the topic in my future research.

Finally, we present the following policy implications to make short- and long-term investment strategies for investors and fund managers according to the above findings. Firstly, in the pre-pandemic period, they can use the return of coffee to predict the return of corn, oats, and soybean oil. Conversely, in the post-pandemic period, they can use the return of soybean oil to predict the return of wheat and oats. The reasons are that, in Fig 3(a), there exist three positive return spillovers from coffee to corn, oats, and soybean oil but, in Fig 3(b) there exist two negative return spillovers from soybean oil to wheat and oats. Moreover, as reported in the data in Table 2, we further presume that, in the pre-pandemic period, a 100% increase in coffee return today causes a 2.93%, 3.84%, and 2.42% rise in return for the corn, oats, and soybean oil tomorrow, respectively. Conversely, in the post-pandemic period, a 100% increase in soybean oil return today causes a 13.4% and 19.27% drop in return for the wheat and oats tomorrow, respectively. This indicates that we can make the following short-term investment strategies to earn more profit. During the pre-pandemic period, if the price of coffee rises today then long traders can buy and hold the position of the corn, oats, or soybean oil today until tomorrow. Conversely, during the post-pandemic period, if the price of soybean oil rises today then short traders can sell the position of the wheat or oats today until tomorrow.

Secondly, in the pre-pandemic period, they can employ the volatility of oats to forecast the volatility of wheat, soybean, and coffee. Conversely, in the post-pandemic period, the volatility of oats is used to predict the volatility of wheat, corn, soybean, soybean oil, and coffee. The reasons are that, in Fig 4(a), there exist three positive volatility spillovers from oat to wheat, soybean, and coffee but, in Fig 4(b) there exist five positive volatility spillovers from oat to wheat, corn, soybean, soybean oil, and coffee. Moreover, as reported in the data in Table 3, we further presume that, in the pre-pandemic period, a 100% decrease in oats volatility today causes a 1.93%, 2.01%, and 5.86% drop in volatility for the wheat, soybean, and coffee tomorrow, respectively. Conversely, in the post-pandemic period, a 100% decrease in oats volatility today causes a 3.49%, 2.64%, 3.05%, 0.74%, and 7.90% drop in volatility for the wheat, corn, soybean, soybean oil, and coffee tomorrow, respectively. This indicates that we can make the following short-term investment strategies to reduce the risk. During the pre-pandemic period, if the volatility of oats decreases today then long traders can buy and hold the position of the wheat, soybean, or coffee today until tomorrow. Conversely, during the post-pandemic period, if the volatility of oats decreases today then long traders can buy and hold the position of the wheat, corn, soybean, soybean oil, or coffee today until tomorrow.

Thirdly, for the pre-pandemic and post-pandemic periods, they can use sugar to hedge the return of oats and the risk of soybeans in the short-term period because soybeans and oats are the maximum net receivers in the volatility and return spillover network for two subperiods, respectively. In addition, sugar is the least active asset in the volatility and return spillover networks for two subperiods. Fourthly, they can select coffee to achieve risk diversification for portfolios in the agricultural market because coffee is the least active asset in the correlation network for two subperiods as shown in Fig 5(a) and 5(b). Moreover, as reported in the data in Table 4, we find that, in the pre-pandemic period, the cr-cf (0.0123), sb-cf (0.0132), and so-cf (0.0022) pair of data respectively have the smallest covariance value within the six corn-based, six soybean-based, and six soybean oil-based pairs of data. This indicates that we can make the following long-term investment strategy to reduce the risk of portfolio in the agricultural market. During the pre-pandemic period, if a portfolio includes corn, soybean, or soybean oil,

then the investors can add coffee to reduce the risk for this portfolio because the value of the portfolio's risk is proportion to the value of covariance. A similar long-term investment strategy also is made in the post-pandemic period.

## Supporting information

**S1 Data.**
(ZIP)

## Acknowledgments

Authors are grateful to the editor and anonymous reviewers for providing suggestions which helped in improving the quality of the manuscript.

## Author Contributions

**Conceptualization:** Jung-Bin Su.

**Data curation:** Jung-Bin Su.

**Formal analysis:** Jung-Bin Su.

**Funding acquisition:** Jung-Bin Su.

**Investigation:** Jung-Bin Su.

**Methodology:** Jung-Bin Su.

**Project administration:** Jung-Bin Su.

**Resources:** Jung-Bin Su.

**Software:** Jung-Bin Su.

**Supervision:** Jung-Bin Su.

**Validation:** Jung-Bin Su.

**Visualization:** Jung-Bin Su.

**Writing – original draft:** Jung-Bin Su.

**Writing – review & editing:** Jung-Bin Su.

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
