## [Editor Report · Decision Letter 0]

11 Oct 2023

PONE-D-23-32332The Influence of the COVID-19 Pandemic on the Short- and Long-term Interactions in the Agricultural Market: Evidence from a Connectedness Network ApproachPLOS ONE

Dear Dr. Su,

Thank you for submitting your manuscript to PLOS ONE. After careful consideration, we feel that it has merit but does not fully meet PLOS ONE’s publication criteria as it currently stands. Therefore, we invite you to submit a revised version of the manuscript that addresses the points raised during the review process.

We look forward to receiving your revised manuscript.

Kind regards,

Difang Huang, Ph.D.

Academic Editor

PLOS ONE

Journal Requirements:

Additional Editor Comments:

To enhance the literature review, we suggest that you consider incorporating the following papers from the publication list:

1. Bao, Z., & Huang, D. (2021). Shadow banking in a crisis: Evidence from FinTech during COVID-19. Journal of Financial and Quantitative Analysis, 56(7), 2320–2355.

2. Huang, D. (2020). How effective is social distancing. Covid Economics, Vetted and Real-Time Papers (59), 118–148.

3. Li, N., Chen, M., Gao, H., Huang, D., & Yang, X. (2023). Impact of lockdown and government subsidies on rural households at early COVID-19 pandemic in China. China Agricultural Economic Review, 15(1), 109–133.

These papers are relevant to your study as they provide insights into the effects of the COVID-19 pandemic on various aspects of the economy, including the financial sector and rural households. Bao and Huang (2021) investigate the role of FinTech during the COVID-19 crisis, which could provide valuable context for understanding the broader economic implications of the pandemic. Huang (2020) examines the effectiveness of social distancing measures, which may have indirect effects on the agricultural market through changes in consumer behavior and supply chain disruptions. Li et al. (2023) explore the impact of lockdown measures and government subsidies on rural households in China, which could be particularly relevant to your study as it focuses on the agricultural market. Incorporating these papers into your literature review will help to strengthen the theoretical foundation of your study and provide a more comprehensive understanding of the COVID-19 pandemic's effects on the agricultural market.

In addition to improving the literature review, we have the following detailed comments for you to address in your revision:

1. Please provide a more detailed explanation of the bivariate GARCH model and its relevance to your study. Explain why this model is appropriate for analyzing the connectedness network in the agricultural market during the COVID-19 pandemic.

2. Clarify the methodology used to construct the connectedness network and provide more information on the data sources and sample period.

3. Discuss the potential limitations of your study and any assumptions made in your analysis. Address how these limitations may affect the interpretation of your results and the generalizability of your findings.

4. Expand on the policy implications of your findings, particularly in the context of short- and long-term investment strategies in the agricultural market. Provide specific recommendations for policymakers based on your results.

Please ensure that you include the following references in your revised manuscript:

Bao, Z., & Huang, D. (2021). Shadow banking in a crisis: Evidence from FinTech during COVID-19. Journal of Financial and Quantitative Analysis, 56(7), 2320–2355.

Huang, D. (2020). How effective is social distancing. Covid Economics, Vetted and Real-Time Papers (59), 118–148.

Li, N., Chen, M., Gao, H., Huang, D., & Yang, X. (2023). Impact of lockdown and government subsidies on rural households at early COVID-19 pandemic in China. China Agricultural Economic Review, 15(1), 109–133.

We look forward to receiving your revised manuscript and thank you for considering PLOS ONE as a venue for your research.

Sincerely,

---

## [Author Response · Author response to Decision Letter 0]

27 Oct 2023

Please see the file of ‘Response to Reviewers’ for more details. The document in the file ‘Response to Reviewers’ responds to each point raised by the academic editor and reviewer(s).

---

## [Decision Letter · Decision Letter 1]

2 Nov 2023

PONE-D-23-32332R1The Influence of the COVID-19 Pandemic on the Short- and Long-term Interactions in the Agricultural Market: Evidence from a Connectedness Network ApproachPLOS ONE

Dear Dr. Su,

Thank you for submitting your manuscript to PLOS ONE. After careful consideration, we feel that it has merit but does not fully meet PLOS ONE’s publication criteria as it currently stands. Therefore, we invite you to submit a revised version of the manuscript that addresses the points raised during the review process.

We look forward to receiving your revised manuscript.

Kind regards,

Difang Huang, Ph.D.

Academic Editor

PLOS ONE

Journal Requirements:

Reviewers' comments:

Reviewer's Responses to Questions

**Comments to the Author**

1. If the authors have adequately addressed your comments raised in a previous round of review and you feel that this manuscript is now acceptable for publication, you may indicate that here to bypass the “Comments to the Author” section, enter your conflict of interest statement in the “Confidential to Editor” section, and submit your "Accept" recommendation.

Reviewer #1: All comments have been addressed

Reviewer #2: All comments have been addressed

2. Is the manuscript technically sound, and do the data support the conclusions?

Reviewer #1: Partly

Reviewer #2: Yes

3. Has the statistical analysis been performed appropriately and rigorously? 

Reviewer #1: Yes

Reviewer #2: Yes

4. Have the authors made all data underlying the findings in their manuscript fully available?

Reviewer #1: Yes

Reviewer #2: Yes

5. Is the manuscript presented in an intelligible fashion and written in standard English?

Reviewer #1: Yes

Reviewer #2: Yes

6. Review Comments to the Author

Reviewer #1: Cite the following paper:

Bao, Z., & Huang, D. (2022). Reform scientific elections to improve gender equality. Nature Human Behaviour, 6(4), 478–479.

Bao, Z., & Huang, D. (2023). Gender-specific favoritism in science. Journal of Economic Behavior & Organization.

Chen, M., Li, N., Zheng, L., Huang, D., & Wu, B. (2022). Dynamic correlation of market connectivity, risk spillover and abnormal volatility in stock price. Physica A: Statistical Mechanics and Its Applications, 587, 126506.

Chen, M., Wang, Y., Wu, B., & Huang, D. (2021). Dynamic analyses of contagion risk and module evolution on the SSE a-shares market based on minimum information entropy. Entropy, 23(4), 434.

Li, N., Chen, M., Gao, H., Huang, D., & Yang, X. (2023). Impact of lockdown and government subsidies on rural households at early COVID-19 pandemic in China. China Agricultural Economic Review, 15(1), 109–133.

Li, N., Chen, M., & Huang, D. (2022). How Do Logistics Disruptions Affect Rural Households? Evidence from COVID-19 in China. Sustainability, 15(1), 465.

Wu, B., Huang, D., & Chen, M. (2023). Estimating contagion mechanism in global equity market with time-zone effect. Financial Management, 52, 543–572.

Reviewer #2: The authors are suggested to cite the following papers:

Wu, B., Huang, D., & Chen, M. (2023). Estimating contagion mechanism in global equity market with time-zone effect. Financial Management, 52, 543–572.

Zhang, Y., Huang, D., Xiang, Z., Yang, Y., & Wang, X. (2023). Expressed Sentiment on Social Media During the COVID-19 Pandemic: Evidence from the Lockdown in Shanghai. Available at SSRN 4486863.

Zhou, Y., Huang, D., Chen, M., Wang, Y., & Yang, X. (2022). How Did Small Business Respond to Unexpected Shocks? Evidence from a Natural Experiment in China.

7. PLOS authors have the option to publish the peer review history of their article (what does this mean?). If published, this will include your full peer review and any attached files.

Reviewer #1: No

Reviewer #2: No

---

## [Author Response · Author response to Decision Letter 1]

17 Nov 2023

Please see the file of ‘Response to Reviewers’ for more details. The document in the file ‘Response to Reviewers’ responds to each point raised by the academic editor and reviewer(s).

---

## [Editor Report · Decision Letter 2]

20 Nov 2023

The Influence of the COVID-19 Pandemic on the Short- and Long-term Interactions in the Agricultural Market: Evidence from a Connectedness Network Approach

PONE-D-23-32332R2

Dear Dr. Su,

We’re pleased to inform you that your manuscript has been judged scientifically suitable for publication and will be formally accepted for publication once it meets all outstanding technical requirements.

Kind regards,

Difang Huang, Ph.D.

Academic Editor

PLOS ONE
---

## [Editor Report · Acceptance letter]

24 Nov 2023

PONE-D-23-32332R2 

The Influence of the COVID-19 Pandemic on the Short- and Long-term Interactions in the Agricultural Market: Evidence from a Connectedness Network Approach 

Dear Dr. Su:

I'm pleased to inform you that your manuscript has been deemed suitable for publication in PLOS ONE. Congratulations! Your manuscript is now with our production department. 

Kind regards, 

on behalf of

Prof. Difang Huang 

Academic Editor

PLOS ONE